# Logical Argument from Evil and Theism

Andrea Aguti 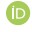

Department of Economics, Society and Politics (DESP), University of Urbino, 61029 Urbino, PU, Italy; andrea.aguti@uniurb.it

**Abstract:** The article argues that the logical argument from evil is dead, and the new version presented by James Sterba cannot resurrect it. In the first part, I say that the logical argument from evil is dead either because, in the version given by Mackie, it was successfully refuted by Plantinga and other theists or because, by inviting a reformulation of theistic doctrines, it was nevertheless superseded by contemporary versions of theism, such as open theism. In the second part, I argue that the two significant moves made by Sterba to resurrect the logical argument from evil fail in their intent either because the premise they start does not necessarily give rise to an atheistic conclusion or because the premise is unacceptable for the theist.

**Keywords:** logical argument from evil; John Mackie; James Sterba; free-will defense; theism

## 1. Premise

Thanks to James Sterba's book *Is a Good God Logically Possible?* (Sterba 2019), the logical argument from evil has once again become the subject of debate in today's philosophy of religion. After its original formulation by John Mackie and its refutation by Alvin Plantinga, the argument seemed, with few exceptions, to have been abandoned and considered dead. Its rebirth, due to Sterba, certainly makes an essential contribution to the debate on theodicy, which is, in my view, the most significant and challenging in the philosophy of religion; however, it is doubtful that this version of argument offers an effective advancement of the discussion in this area.

The thesis I support in this article is that the logical argument from evil is dead, i.e., has become irrelevant. Therefore, the attempt to resurrect it is doomed to fail. Other types of evil arguments, such as evidential arguments, continue to challenge the theist, but not the logical one. The logical argument from evil is dead because, already in Mackie's original version, it has not achieved the purpose of refuting theism. Equally, if its purpose has been to increase the critical awareness of theism, it has already achieved this goal and can be said to be exceeded.

## 2. Why the Logical Argument from Evil Has Died by Not Hitting the Target

To understand why the logical argument from evil is dead, I refer to its original formulation, at least in the contemporary debate on the philosophy of religion, which is that of John Mackie (Mackie 1955). The argument is prima facie an attack on theism because it does not aim to weaken or refute the evidence in favor of the rationality of the latter, as in the case of the arguments of natural atheology, but to point out a logical contradiction within theism. The theist is committed to simultaneously affirming that (1) there is an omnipotent God, (2) this God is perfectly good, and (3) evil exists in the world. However, if God is omnipotent and perfectly good, and the evil in the world is actual, not a mere appearance, the first two propositions exclude the third because, supposedly, an omnipotent and perfectly good God can and wants to create a world where evil does not exist. However, evil exists, so there is no God.

If the argument aims to prove the inconsistency of theism, it must be said that its goal is very ambitious. If so, we should probably see in this argument the expression of the

maximum flourishing of atheistic thought in the contemporary era. Nineteenth-century atheism mainly based its claim on historical verification: the progressive and inevitable replacement of the social functions performed by religion by secular activities, the future solution of the *Welträtsel* by scientific knowledge, cognitive mechanisms that, once brought into the light, would be stuck, feelings and emotions that, once rationalized, would have been overcome. The alternative was the heroic claim of human freedom against God, as in the case of "postulatory atheism".

A form of atheism that has become fully aware of itself; lives in an epistemic context, having no more relevant obstacles; and, therefore, which has become bold, is ready to develop an unum argumentum that attempts to defeat the opponent's field. This is precisely what the logical argument from evil does. As Mackie states, the logical problem raised by the argument is "the problem of clarifying and reconciling a number of beliefs: it is not a scientific problem that might be solved by further observations, or a practical problem that might be solved by a decision or an action" (Mackie 1955, p. 200). If the problem has no solution, as Mackie thinks, the atheist has a lethal weapon in his hand against the opponent. Let us look for a counterpart to this argument in theism. We find it in the so-called ontological argument of Anselm of Canterbury, which, not surprisingly, was formulated in a flourishing era of Christian theological thought. In this case, the belief is that, given a specific definition of God that works as a premise, the argument's conclusion is logically necessary. In both cases, you win or lose; there are no intermediate degrees.

However, philosophers have regularly questioned the certainty of producing a conclusive argument in both cases. This happens because every philosophical argument cannot avoid failure, if by failure we mean, as Peter van Inwagen says, the possibility that the members of an ideal audience, that is, impartial, intellectually honest, and endowed with philosophical and logical acumen (see van Inwagen 2006, p. 42), are not convinced by the argument. This possibility is regularly updated in philosophy, both about theistic and atheistic arguments, representing an invitation to have more modest claims. After the modern critique of natural theology and theodicy, some theists have become too modest, completely renouncing to formulate rational arguments in these areas and therefore yielding to the temptation of fideism, as happens in skeptical theism. Others have taken advantage of this criticism to develop less ambitious strategies without renouncing rational arguments. An attitude of prudence inspired the distinction, which has become classic, between theodicy and defense that arose precisely on the ground of the theistic reply to the logical argument from evil.

This argument stands or falls with the assumption that the theistic God does not have sufficient moral reasons to prevent evil in the world. Still, the theist replies that God, if he exists,[1] has reasons not to avoid evil in the world, although these reasons could not be those the theist thinks he has. In a defense, as Plantinga writes, "the aim is not to say what God's reason is, but at most what God's reason might possibly be" (Plantinga 1977, p. 28). Or, in the words of van Inwagen, the theist offers a story about God that represents "God as having reasons for allowing the existence of evil, reasons that, if the rest of the story were true, would be good ones" (van Inwagen 2006, p. 66). How good these reasons are is something to be evaluated. Still, if there are, they are, in any case, sufficient to remove the logical contradiction between the existence of an omnipotent and perfectly good God and to downgrade the logical argument to an evidential argument. According to the latter, theism is probably false, and faith in God is unreasonable. Thus, an atheist has rational grounds for not believing in God but nothing more. Even an irreducible atheist such as Richard Dawkins must concede that there is a minimal probability that God exists.

Some atheists, such as Graham Oppy, have recognized that the logical argument from evil is dead, but only in Mackie's version, and this does not exclude that, in the future, there may be different versions (Sterba's argument might precisely be one of these that Oppy has in mind) (see Oppy 2017, p. 63). To demonstrate the weakness of Plantinga's free-will defense, that is, his inability to dismiss any logical argument, Oppy proposes a different one that starts from the following premises: "1. If God exists, God is the perfect *ex*

*nihilo* creator of our universe; 2. Our universe is imperfect. 3. The actions of a perfect being cannot decrease the degree of perfection of the world. 4. If God exists, then, prior to all creation, the world is perfect" (Oppy 2017, p. 54).[2] If we accept these premises, the world should be perfect; that is, it should not contain any kind of evil, to the point that, as Oppy writes, "even the slightest toothache is *a prima facie* intellectual problem for perfect-being theists" (Oppy 2017, p. 55). However, of course, the world contains a lot of evils, and from this, considering the third premise, we infer that God does not exist.

However, even this version of the logical argument is not conclusive. It could be that the world in mente Dei is not perfect as Oppy thinks it is. If a perfect world means the best of all possible worlds, then if something like this makes sense (and I think not), it is not sure that it does not contain any evil. More generally, since the world is ontologically different from God, its perfection is necessarily inferior to the divine one. Compared with the latter, the world is constitutively imperfect; that is what, from Augustine to Leibniz, has been called "metaphysical evil". In sum, a perfect but real world will never be as perfect as an ideal world which is part of divine perfection, so Oppy's third premise fails. However, even if we recognize the validity of all the premises set by Oppy, they do not necessarily deduce the non-existence of God, but only that, in the passage from the intellect to reality achieved with creation, something went wrong. In this case, the argument would strike not theism but only a version of theism, i.e., perfect-being theism. The reproach to God, in this case, would be that of having created an imperfect world, not that of existing.

Thus, even this reformulation of the logical argument from evil, like the others, fails in the sense pointed out by van Inwagen. Of course, I repeat, the failure of this argument does not mean that the problem of evil does not continue to present a formidable challenge for the theist, nor does it mean that free-will defense helps respond convincingly to all kinds of evil in the world. Michael Tooley argues that a defender of what he calls an "incompatibility argument from evil" can always render free-will defenses irrelevant "by formulating an argument from evil in terms of natural evils" (Tooley 2019, p. 6). Indeed, natural evils, the suffering of beasts, and even the suffering of human beings, who, for various reasons, cannot fully exercise their free will, are not covered by free-will defenses, or they are not in a way that seems plausible. Precisely for this reason, in her treatment of the problem of theodicy, Eleonore Stump took up the free-will defense, delimiting it "to the suffering of mentally fully functional adult human beings" (Stump 2010, p. 5). This delimitation can leave you unsatisfied, but if the defense works in this case, it shows that God has reasons for allowing evil and suffering in the world. This is sufficient to reject the logical argument from evil. If God has reasons for allowing the suffering of human beings, it is plausible to think that he also has reasons for allowing that of animals or humans who are not mentally fully functional.

## 3. Why the Logical Argument from Evil Has Died Hitting the Target

The logical argument from evil in Mackie's version is an argument against theism, that is, as an argument that refutes the existence of an omnipotent and perfectly good God. However, let us try to consider it as an argument that does not refute theism but invites the theist to understand better what it means to speak of an omnipotent and perfectly good God and have faith in him without falling into contradiction. If so, we would be faced with one of those cases in which atheistic arguments serve to purify theistic faith. In this sense, perhaps, we might interpret it in meliorem partem, like Hume's arguments against natural theology in his *Dialogues Concerning Natural Religion* or the argument against miracles in his *Enquiry Concerning Human Understanding*. They do not dismiss theism, understood as a belief in an invisible and intelligent power that orders the course of nature (this is Hume's idea of theism, an idea undoubtedly reductive for most theists); they free it from superstition or inconsistent doctrines.

Since in Mackie's version of the logical argument, the prevailing focus is on the attribute of omnipotence, I limit myself to a few considerations on this topic. The free-will defense works only on the assumption that God cannot control the will of human beings.

This gives rise, as is known, to what Mackie calls "the paradox of omnipotence", which arises from the following question: "Can an omnipotent being make a thing which he cannot subsequently control?" (Mackie 1955, p. 210). In his book *The Miracle of Theism*, he asserts that it is an "undecidable question" (Mackie 1982, p. 161). The notion of omnipotence implies that one can have control over anything, so its denial is self-contradictory. Still, at the same time, it is equally impossible to admit that God can control what he, as an omnipotent being, has made uncontrollable.

However, the impossibility of resolving the paradox of omnipotence logically does not present any significant objection to theism. As Peter Geach has made clear (see Geach 1977, pp. 3ff.), a theist may think that, intuitively, there are many things that God cannot do, such as telling lies or not keeping his promises, while agreeing with the fact that there is no coherent solution to the paradox of omnipotence. In doing this, a theist shows that he does not need to believe in God's omnipotence in the sense that "God can do everything", but only in the sense that God is almighty, that is, that he has a providential plan about human beings that can never be frustrated. Some might object that God wants the salvation of all men (according to *1 Tim* 2, 4) and that one of the outcomes of the free-will defense is that not everyone will be saved, thus frustrating God's will. Still, the meaning of the biblical phrase is that God wants to save all those men who want to be saved, a sense that is precisely consistent with the free-will defense.

This brings us back to the core of Mackie's objection to the free-will defense. It does not object to the idea that God cannot force human beings to choose the good but that he has not created a world where humans "always freely choose the good" (Mackie 1982, p. 164). I think that this objection must be taken seriously by a theist. There is nothing logically contradictory in assuming that all human beings can always freely choose the good; as Mackie observes, it is what Christian utopians hope will one day happen. Plantinga's thesis about Transworld Depravity, for which "every world God can actualize is such that" the man "is significantly free in it, he takes at least one wrong action" (Plantinga 1977, p. 47), is plausible if one considers the postlapsarian state, but not the prelapsarian state. Perhaps God created the world by offering the first human beings the possibility of always freely choosing the good. Still, they simply did not do it, and the initial error resulted in the impossibility of doing so throughout the history of humanity. Did God know that they would not do it? According to Mackie, the free-will defense works fully if God does not know future contingents because, by creating free beings, he cannot make them so that he always knows what they will freely choose. This move, however, would not be painless for the theist because, as Mackie observes, it would lead him to have a minor conception of omniscience and therefore also of divine omnipotence, "to put God very firmly inside time", contrary to the ordinary religious view of God's eternity, and, considering that the world could even be worse than it is, to make God run a great risk, exposing him "to a charge of gross negligence or recklessness". Nonetheless, as Mackie acknowledges, "there may be some way of adjusting these [doctrines] which avoids an internal contradiction without giving up anything essential to theism". However, he adds, "none has yet been clearly presented, and there is a strong presumption that theism cannot be made coherent without a serious change in at least one of its central doctrines" (Mackie 1982, p. 176).

From the time Mackie made these considerations, theists have taken the path he suggested seriously, as representatives of open theism well demonstrate. For example, William Hasker is willing to acknowledge that God took a risk by creating the world. In the conclusion of his book *God, Time and Knowledge*, he states that "the best Christian theodicy will deny middle knowledge and will affirm forcefully that *God the Creator and Redeemer is a risk taker!*" (Hasker 1989, p. 205). Nevertheless, I wonder if the idea of a God who runs risks is appropriate for the theistic God. In my view, refuting God's omniscience about the future mainly serves to avoid inconsistencies regarding this complicated issue. One of these consists, as Geach observes, in the idea common to classical theism that God sees future events as they are in themselves, evidently based on the assumption that the future "exists" already. The future, however, does not "exist" already, in a sense expressed in logic

by the existential quantifier, but consists "of certain actual trends and tendencies in the present that have not yet been fulfilled" (Geach 1977, p. 53). To affirm that God sees the future as already present is contradictory because the future, by definition, is not present. Attributing this ability to God does not magically make the contradiction disappear.

Denying that God sees the future as present does not mean denying that God knows the future because, as Geach still observes, "God knows the future by controlling it" (Geach 1977, p. 57). In other terms, the future's divine knowledge is a function of God's omnipotence. The image proposed by Geach of the great chess master who has everything under control and who, once established to checkmate, no one can force to improvise, is perhaps not entirely appropriate because if God does not know in advance the moves of his adversaries, he cannot even plan his moves. Maybe he will have to improvise. However, the aim of the image is precise: God's final intentions cannot be frustrated, and what he has established, he always obtains. Therefore, for a theist, abandoning omnipotence in the sense that "God can do anything" does not mean abandoning his almightiness.[3]

If such a conception is plausible, theists have managed to respond coherently, using free-will defense, to the logical argument from evil. Thus, it can be overcome after having fulfilled its critical function. It has highlighted an apparent inconsistency of theism. Still, theists, through a reformulation of their theses, have overcome the objection precisely as they suppose that evil is destined to be overcome by good. Even in this perspective, therefore, the logical argument from theism is dead.

## 4. Sterba's Logical Argument from Evil

Having established this, what about the logical argument proposed by Sterba? Since the publication of his book, a considerable discussion has developed on it, which has seen many scholars intervene and Sterba himself replying. There is no need to repeat Sterba's arguments, the objections raised, Sterba's rejoinder. I would just like to make some considerations on two specific points.

First point: Sterba considers the logical argument from evil conclusive for the option in favor of atheism. He does it in general, but also personally. In a summary article of his book, he recalls that he was not an atheist until he formulated, in recent years, his version of the argument from evil and states that "my commitment to atheism is only as strong as the soundness and validity of my argument. Undercut my argument and poof, at least in my case, no more atheist" (Sterba 2020, p. 203). This statement indeed accounts for the intellectual honesty of its author and the non-dogmatic nature of his commitment to atheism. Nonetheless, making this commitment dependent on a single argument seems reckless because, as I previously said, every philosophical argument is open to relevant objections and can be fully convincing or not. However, there is another reason: I think that a well-founded option for theism or atheism should spring from a more comprehensive epistemic attitude. To be an atheist means having the conviction that theism is not the best possible explanation for the problem of the origin of the world, for the apparent design of living beings, for the issue of human nature, of the meaning of life, of the foundation of morality, of life after death, for the existence of the tremendous amount of religious experience present in the world, of miracles, etc. Mackie's book *The Miracle of Theism*, which replies to Richard Swinburne's cumulative argument for theism in *The Existence of God*, shows such an epistemic attitude. In this context, a single argument can be more robust and give a greater impetus to tip the scales on one side rather than the other. Still, none alone is enough, and, eventually, it is easier to refute a single argument than a series of converging arguments. Naturally, Sterba, to date, has not evaluated the objections raised to him as capable of finding a flaw in his argument. For this reason, he is entitled to consider it valid and to remain an atheist. Still, perhaps he will accept the invitation to reflect further if an option so existentially demanding, like the atheistic one, can be based on a single argument, however suitable it may be.

In this regard, it is well to add something else: let us admit that the logical argument from evil in Sterba's version is successful and immune from flaws. Not for this, atheism

would be its logical consequence. The most consistent and frequent meaning of the term "atheist" is someone who denies God or divinities exist (see Oppy 2018, p. 3). Let us admit that the logical argument from evil makes faith in an almighty and perfectly good God inconsistent. Still, it leaves the possibility of believing in a God who does not possess one of these two properties. The finite God of John S. Mill, Max Scheler, Hans Jonas, or contemporary panentheism is a feasible option for those who look at the problem of evil as an insurmountable obstacle to believing in theistic God. According to the distinction proposed by Rowe, in the latter case, we would speak of a theism "in a broad sense" (Rowe 1979, p. 335). At most, we would talk about "implicit atheism", as does Italian philosopher Cornelio Fabro (see Fabro 2013, p. 84) but not of "atheism" sic et simpliciter.

In the conclusion of his book, Sterba rejects this possibility, stating that such a God "would have to be extremely immoral or extremely weak" and "no useful purpose would be served by hypothesizing such a limited god who would either *be so much more evil* than all our greatest villains or, while moral, would *be so much less powerful than ourselves*" (Sterba 2019, p. 192). Here it seems that Sterba shares with the theist the idea that an impotent or evil god is not "God" in the real sense.

However, the greater rational coherence of a God with all the perfections does not exclude the existence of minor divinities. From the Anselmian argument, if it works, you derive the logical necessity to affirm the existence of "God", but not the non-existence of "god". If the only possible alternative were between theism and atheism, in the latter, we should include all conceptions of the divine other than theistic ones. However, this would have the consequence that religious views other than monotheistic ones should all be considered expressions of atheism, a contradictory consequence. Are they not religions precisely because they have some concept of the divine and worship a deity? The accusation of atheism can naturally be launched against those who have a conception of divinity other than the one held to be true (as the pagans did in ancient times toward Christians or as Christian theologians did in modern times against Spinoza or Fichte in the *Atheismustreit*). Still, those affected by this accusation can rightly reply that they are not atheists because there is no single concept of god, and one can mean different things with the term "god". In short, the logical argument from evil can lead to atheism, but it does not necessarily do it. For being consistently atheist, there is a need to formulate not only objections to the existence of God but also to "gods" and, more generally, to the rational plausibility of a religious worldview.

So, we come to the second point: the novelty of Sterba's logical argument from evil consists of its reformulation in moral terms. More precisely, it highlights that free-will defense does not work if one accepts a morally qualified concept of freedom and asserts the so-called Pauline Principle as a set of moral obligations to which a God with all perfections would be subject. I believe that neither of these moves can resurrect the logical argument from evil, but the reasons for failure differ. In the first case, Sterba's move may be shared by the theist, and his defense may be reformulated successfully in terms of what Sterba himself calls "Greater Moral Good Defense" (Sterba 2019, p. 30); in the second case, instead, a theist must reject the premise of Sterba's reasoning, declaring its irrelevance.

Let us start with the first move. Sterba states the difference from Plantinga's conception of freedom as follows: "For me significant freedoms are those freedoms a just political state would want to protect since that would fairly secure each person's fundamental interests" (Sterba 2019, p. 12). While freedom for Plantinga indicates the ability to perform or refrain from a morally significant action, for Sterba, freedom is linked to a sphere of interests or rights that an ideal political state should preserve and which, in analogy with the latter, God should maintain too. The preservation of these interests or rights implies the practice of constraining the freedom of those who do injustice, which a just state does regularly, even if insufficiently, but which God does not seem to do, as evidenced by the presence of horrendous evils in the world. While God's non-interference with man's freedom is justified when dealing with lesser evils, this justification falls in the face of horrendous evils. Thus, as Sterba observes, the problem with theodicy is not that God creates us free

but that he "fails to restrict the lesser freedoms of wrongdoers to secure the more significant freedoms of their victims" (Sterba 2019, p. 29).

It seems to me that a theist can follow this line of reasoning without accepting its conclusions. A theist, that is, can agree that the free-will defense alone is not enough to face the objection based on the existence of horrendous evils because, if considered in isolation, it can give rise to a misconception that Sterba's remarks help to highlight. The misconception consists in thinking that any interference of God with human freedom, especially with the consequences of freely chosen actions, consists of negating the latter. However, this is not the case because, as Sterba rightly observes, "God can also promote freedom ( … ) by actually interfering with the freedom of some of our free actions at certain times" (Sterba 2019, p. 27). Therefore, a theist should not think that the price to pay for putting the responsibility for moral evil on human shoulders is to keep God out of the game. If we admit the logical possibility of this interference, as Sterba does against Plantinga, from it one can presume, against Sterba, that God actually interferes with human freedom and that he does so with extraordinary interventions, as happens in miracles, or in an ordinary way, through worldly causality. The presence of horrendous evils does not constitute an objection to the principle of God's non-interference. In the face of horrendous evils, a theist can only acknowledge that what one would have expected, interference from God, did not happen, not that God does not exist, or that God never interferes in human affairs.

The question that horrendous evils pose to the theist is not why God does not intervene in general but why God has not intervened in these cases. From a theistic point of view, I think there is only one plausible answer to this question: God permits horrendous evil with the aim of a greater good. This response, which is that of classical Christian theodicy, like that of Thomas Aquinas, supposes that suffering is a means to obtain goods that otherwise would not be possible. Spiritual goods in this life, the good of beatitude in the ultramundane life. Ultimately, this response denies that there are horrendous evils, that is, that there is suffering without a teleological orientation to good.

You may say that the answer is wrong because, by definition, horrendous evils are such precisely because they deny this orientation. Still, the problem lies precisely in the judgment that we express on these evils without having sufficient evidence to do so. Suppose evils are permitted to obtain spiritual goods. In that case, these are less visible than material ones, and the connection between suffering and good is often hidden. If evils are permitted to obtain the supreme good of beatitude, this good is an object of faith in this life, not of vision. In these cases, a theist has no evidence to say that evil is not absurd, but he can certainly assume that it is not if an almighty and good God exists. In this conception, the only horrendous evils are those that the wrongdoers experience and will experience as the fruit of their actions. Still, they are not even absurd because they represent the punishment consequent to their guilt. It will be noted that this response is different from that of skeptical theism, toward which Sterba shows justified perplexities: in skeptical theism, God's reasons for allowing evil remain unknown to us, while in our case, God's reasons may be, at least partially, known to the human being.

Assuming this point, we come to Sterba's second move, based on the Pauline Principle. According to this principle, it is not permissible to do evil to obtain good, whatever it may be. A "Greater Moral Good Defense" seems challenged by this principle, at least in the case of horrendous evils, because trivial or easily repairable evils are an obvious exception. As I said earlier, I think Sterba's line of reasoning must be rejected entirely on this point. The Pauline Principle prohibits doing evil to obtain good, but it does not prohibit allowing evil if this permission is the only way to prevent a greater evil.[4] The doctrine of the Double Effect, which relates to the Pauline Principle, explains this point, with the only difficulty in admitting that God did not foresee in detail the unwanted effects of his permission. In any case, whether God has foreseen or not foreseen such effects in detail, it remains a strong point of theism that God always wants the good and that nothing can frustrate his will, even when it is made explicit through permission of evil. The distinction of Thomas Aquinas

between antecedent and consequent can help bring this point into focus. According to this distinction, the permission of what God does not want in his antecedent will, that is, abstracting from actual circumstances, is a good thing given these circumstances and is therefore willed according to his consequent will (see Stump 2003, pp. 458 ff.). So, to give an example relevant to our problem, the choice of doing evil by one of his creatures is not willed by God according to his antecedent will but is permitted according to his consequent will not to destroy his freedom. It is true that permission, as Sterba observes, "is always an intentional act" (Sterba 2019, p. 123). Still, it is not the intentionality of the act that makes someone guilty if circumstances make it the right thing to do or the only possible thing to do.

The Pauline Principle, in Sterba's formulation, on the other hand, takes the form of three moral evil-prevention requirements (see Sterba 2019, pp. 151 ff.), which result in moral obligations that God should satisfy in analogy to what an ideally just political state does. However, the idea that God is subject to moral obligations is inadmissible for a theist. In chapter VI of his book, Sterba confronts Brian Davies's negation that God is a moral agent. However, I think that Davies' thesis is not entirely representative of the theistic conception of morality because it underlies a theological apophatism that a theist may not share.

The whole question, in my opinion, should be considered as follows: God has no moral obligation, but, by his nature, he cannot do certain things that are morally significant. God, for example, cannot lie or want to do evil in the sense of the antecedent will. Thus, God is not obliged to create the world, and creation is a supererogatory act. The creation of the world and God's providential plan, however, imply the creation of a physical order and a moral order that are in a close relationship, an order that is valid for human beings but not for God, who is its creator. God can deviate from the physical order by working miracles, that is, events that exceed or violate the causal powers of things, not their nature, and he can deviate from the moral order by commanding acts that are contrary to it, as in the case of the sacrifice of Isaac from part of Abraham. The power to command actions that violate the moral law shows the sovereignty of God, that is, of the Legislator, over the latter. This point is, it seems to me, what must be conceded to a divine command theory, but without opposing the latter to natural law theory.

The element that allows us to keep these two theories together, which are different but not necessarily opposite,[5] is that both natural moral law and divine command are aimed at the good or the greater good. In this sense, God can be conceived as an ideal moral agent who always acts for the good, even when it seems to us that this is not the case. A theist cannot consistently believe that God violates the moral law arbitrarily or just to demonstrate his power, nor is he forced to believe that God's moral action completely differs from any human moral standard. The ontological difference between God and human beings justifies only a certain degree of agnosticism about what matters to us as good; God knows thoroughly what is good for us, and this knowledge justifies him in allowing evil and suffering and commanding an action contrary to the moral law. In any case, this agnosticism rests on the firm conviction that everything God does is for our good and that his will cannot be thwarted.

The idea that God has moral obligations to satisfy reveals, in my opinion, an anthropomorphic attitude toward God which ultimately produces a misconception of his nature. An ideal political state made up of human beings is undoubtedly subject to moral obligations, and human beings with superpowers (superheroes) are equally so, but God, who is the creator of everything and therefore also of moral obligations, is not. For this, I conclude that Sterba's argument, based on the Pauline Principle, builds on a premise that the theist cannot accept.

**Funding:** This research received no external funding.

**Conflicts of Interest:** The author declares no conflict of interest.

## Notes

[1] On this point, a theist can adopt what W. Rowe has called "the G. E. Moore shift", which consists of assuming as a premise the negation of the opposing argument's conclusion and drawing the negation of one of the premises of the opposing argument. In other words, for the atheist, God does not exist, but if God exists, a premise that must be acquired through arguments other than those used in the theodicy, then certainly God has morally sufficient reasons to allow evil (see Rowe 1979, p. 339).

[2] In the list of Oppy's premises, I have substituted numbers for letters.

[3] Ultimately, this conception is compatible with Hasker's view: "God knows, to be sure, that evils will occur, but for the most part he will not have specifically decreed or incorporated into his plan for the world the particular instances of evil which actually occur. And this opens up for us the possibility of attributing to God certain general strategies by which he governs the world, strategies which are, as a whole, ordered for the good of the creation, but whose detailed consequences are not foreseen or intended by God prior to the decision to adopt them" (Hasker 2004, p. 118). The difference with classical theism is that this conception admits the existence of evils in the world that are not compensated, at least in this life, by a greater good.

[4] Among others, Almeida has pointed out this issue in his review of Sterba 2019 (Almeida 2020, p. 248).

[5] Aquinas' ethical thought is often understood in the light of the natural moral law's theory. Still, it contains many elements consistent with a theory of divine commands (see Clanton and Martin 2019).

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
