# Peer review of "Logical Argument from Evil and Theism"

_religions, doi:10.3390/rel13111007_

Round 1

Reviewer 1 Report

The first half of this paper is an interesting discussion of the purpose and general failings of logical problems of evil. The discussion of Geach was interesting and relevant to these debates. I don't think this has yet come out in the published articles in this journal and it is on this basis that I think this paper makes a sufficient contribution to merit publication.

Discussion of Sterba's own version of the logical problem is restricted to the second half of the paper. Reasons are given for rejecting this argument too, though these reasons have been developed by a number of other articles already recently published in the journal.

I noticed a few spelling mistakes, unfortunately I lost the copy where I had them highlighted. Please have a close read to find them. One was in footnote 4 on pg 8, where 'ha' should be 'has'.

Author Response

I corrected the typo indicated by the reviewer, along with others found in the original version of my article.

Reviewer 2 Report

The author of the article defends the thesis that the argument from evil is dead, contrary to the view of James Sterba. It is dead because it does not refute theism.

First, the Author recalls the original formulation of the logical argument by John Mackie. He points out that the purpose of the logical argument is to show that the following three sentences cannot be held true simultaneously: 1) there is an omnipotent God; 2) this God is perfectly good; 3) evil exists in the world. Since evil is an empirical fact, it means that God does not exist. The author also mentions other versions of the same argument. The author criticizes and shows the weaknesses of all these versions. He emphasizes that it can be considered as an argument that does not refute theism, but invites the theist to understand better what it means to speak of an omnipotent and perfectly good God and have faith in him without falling into contradiction.

Then the author presents the problem of God's omniscience and omnipotence, as well as attempts to solve this problem by various authors. Up to this point, the work is imitative. The conclusion is that the limitation of God's omnipotence weakens the logical argument from evil.

The next step is to present the Sterba's version of logical argument from evil. Unfortunately, the author quotes it too vaguely. The unfamiliar reader does not learn much from this presentation.

Criticizing Sterba's approach, the author claims that “a well-founded option for theism or atheism should spring from a more comprehensive epistemic attitude. To be an atheist means having the conviction that theism is not the best possible explanation for the problem of the origin of the world, for the apparent finalism of living beings, for the issue of human nature, of the meaning of life, of the foundation of morality etc. “.

The Author claims: “Let's admit that the logical argument from evil makes faith in an almighty and perfectly good God inconsistent. Still, it leaves the possibility of believing in a God who does not possess one of these two properties ”(lines 254-257). Personally, it is difficult for me to agree with this line of defence of theism, since the notion of God not almighty or morally imperfect does not belong to the classical concept of the absolute. This thought is also confirmed by Sterba (lines 263-268). However, the author rightly points out that there are such positions.

It is also difficult to agree that the greater rational coherence of a God with all the perfections does not exclude the existence of minor divinities. The author believes that, if Anselm's argument works, God is the subject of all perfections. But there may also be “gods” who do not have all the perfections. I wonder if the lack of all perfections is not a reason that they are not God/“gods”. People and sheep might as well be said to be “gods” because humans and sheep also do not have all the perfections. Being God—at least in the concept of Anselm (or Descartes)—implies having all the perfections. There is nothing between theism and atheism. Polytheism is not equal to atheism. I disagree with the statement that the argument from evil can lead to atheism, but it does not do it necessarily. If the logical argument from evil were right (I'm not saying it is), it would have to lead to atheism.

Then the Author claims that the novelty of Sterba's logical argument from evil consists of its reformulation in moral terms. The author believes that this move cannot resurrect the logical argument from evil. Sterba observes that the problem with theodicy is not that God creates us free, but that he “fails to restrict the lesser freedoms of wrongdoers to secure the more significant freedoms of their victims”. According to the Author, the misconception consists in thinking that any interference of God with human freedom, especially with the consequences of freely chosen actions, consists of negating the latter. In the face of horrendous evils, a theist can only acknowledge that what one would have expected, interference from God, did not happen, not that God does not exist, or that God never interferes in human affairs.

Ultimately, the author solves the problem in the spirit of classical theodicy (in the Christian version). He concludes that Sterba's argument, based on the Pauline Principle, moves from a premise that the theist cannot accept.

The article is interesting because it shows the argument from evil from different perspectives. It is sometimes difficult to distinguish the views of the Author from the views of other philosophers. It could be improved. I recommend the text for publication, after making minor corrections.

Author Response

From line 257 to line 291, I try to give reasons why an argument against perfect-being theism is not an argument against theism in general. To justify this view, I refer to W. Rowe's distinction between theism in a "narrow sense" and theism in a "broader sense". It seems to me a well-founded distinction both on the philosophical and religious level. Open theism differs from perfect-being theism, but it's still theism. As the reviewer notes, polytheism is not equal to atheism. 

In the conclusion of the article, my personal view emerges quite clearly. 

Reviewer 3 Report

Comments on “Logical Arguments from Evil and Theism”

 1. Line 22 - the claim that the Logical Problem of Evil (LPOE) is “dead” does this mean unsound or irrelevant?

2. Line 26 – does “critical awareness” = make improbable?

3. Typo in Line 59.

4. Note that Peter van Inwagen uses lower case for the “van”. Lines 66, 81, 117, and possibly elsewhere.

5. Line 70 needs work, not grammatically correct.

6. Line 87, consider Rowe’s “friendly atheism” as regards the evidential argument. He holds that a proponent of the EPOE can still hold that a theist can be rational (though incorrect).

7. Line 92, drop the “one” after “Sterba’s”.

8. Line 94 and following – the discussion ignores the Principle of Organic Unities – the idea that a bad part could enhance the overall value of the whole.

9. Line 97 – “world” and not “word”.

10. The argument in line 103 and following employs “world” ambiguously.  The argument seems to vacillate between “possible world” and “the created universe”.

11. Lines 126 and following – couldn’t an LPOE be formulated that evades Stump’s minimal FWD? Isn’t that what Sterba is trying with animal suffering?

12. Line 135 – it seems odd to say that Mackie (or Mackie’s argument) “invites the theist” to understand better theism.

13. Line 145 and elsewhere – probably best to place the definite article before FWD (the FWD).

14. Line 217 – the idea of possible inconsistency.  P is possibly inconsistent -  If the modality is logical, then P is inconsistent. The modality must be epistemic – we are not sure if P is inconsistent.

15. Lines 62-70, 235 – the employment of PVI’s idea of philosophical failure – “every philosophical argument cannot avoid failure” is open to relevant objections” (cannot be conclusive, or free from relevant objection). Is there a threat of self-referential undermining?  This comment may not be salient to the point of the paper.

16. Line 239 – explain “finalism”. May not be known to the reader.

17. line 257 – “finite” might be a better choice than “impotent”.

18. Lines 336-37 – this point seems to overlook animal suffering in the evolutionary history prior to human evolution.  

19. Lines 383 & 385 – maybe “deviate” rather than “derogate”?

20. Line 394 – “acts” rather than “asks”.

21. Line 407 – maybe “builds on”  rather than “moves from”.

Author Response

This review was very useful. I've incorporated into the revised version of my article all the indications about typos and language improvements.

I've tried to clarify all the ambiguous points indicated, except point 8, line 94, and point 10, line 103. In these cases, I'm criticizing Oppy's version of the logical argument from evil. It seems to me that Oppy's argument is ambiguous, not my criticism.

Round 2

Reviewer 3 Report

None.